# Predicting central choroidal thickness from colour fundus photographs using deep learning

Yusuke Arai[1], Hidenori Takahashi[1]*, Takuya Takayama[1], Siamak Yousefi[2,3], Hironobu Tampo[1], Takehiro Yamashita[4], Tetsuya Hasegawa[5], Tomohiro Ohgami[6], Shozo Sonoda[4], Yoshiaki Tanaka[5], Satoru Inoda[1], Shinichi Sakamoto[1], Hidetoshi Kawashima[1], Yasuo Yanagi[7,8]

1 Department of Ophthalmology, Jichi Medical University, Shimotsuke, Tochigi, Japan, 2 Department of Ophthalmology, University of Tennessee Health Science Center, Memphis, Tennessee, United States of America, 3 Department of Genetics, Genomics, and Informatics, University of Tennessee Health Science Center, Memphis, Tennessee, United States of America, 4 Department of Ophthalmology, Kagoshima University, Kagoshima, Japan, 5 Department of Ophthalmology, Saitama Medical Center, Jichi Medical University, Saitama, Japan, 6 Department of Ophthalmology, Ibaraki Seinan Medical Center, Ibaraki, Japan, 7 Department of Ophthalmology, Yokohama City University, Kanagawa, Japan, 8 Medical Retina, Singapore Eye Research Institute, Singapore, Singapore

* takahah-tky@umin.ac.jp

**Data Availability Statement:** All relevant data are available from the figshare database (DOI https://doi.org/10.6084/m9.figshare.9581045.v1).

## Abstract

The estimation of central choroidal thickness from colour fundus images can improve disease detection. We developed a deep learning method to estimate central choroidal thickness from colour fundus images at a single institution, using independent datasets from other institutions for validation. A total of 2,548 images from patients who underwent same-day optical coherence tomography examination and colour fundus imaging at the outpatient clinic of Jichi Medical University Hospital were retrospectively analysed. For validation, 393 images from three institutions were used. Patients with signs of subretinal haemorrhage, central serous detachment, retinal pigment epithelial detachment, and/or macular oedema were excluded. All other fundus photographs with a visible pigment epithelium were included. The main outcome measure was the standard deviation of 10-fold cross-validation. Validation was performed using the original algorithm and the algorithm after learning based on images from all institutions. The standard deviation of 10-fold cross-validation was 73 μm. The standard deviation for other institutions was reduced by re-learning. We describe the first application and validation of a deep learning approach for the estimation of central choroidal thickness from fundus images. This algorithm is expected to help graders judge choroidal thickening and thinning.

## Introduction

Interest in choroidal thickness has grown since the development of enhanced depth imaging optical coherence tomography (EDI-OCT) for qualitative measurements [1]. An increase in

**Funding:** The author(s) received no specific funding for this work.

**Competing interests:** Dr Arai, Dr Takayama, Dr Yousefi, Dr Tampo, Dr Yamashita, Dr Hasegawa, Dr Ohgami, Dr Sonoda, Dr Tanaka, Dr Inoda, and Dr Sakamoto declare no potential conflict of interest. Dr Takahashi: Lecturer's fees from Novartis Pharmaceuticals, Kowa Pharmaceutical, Senju Pharmaceutical, Alcon Pharmaceuticals, Santen Pharmaceutical, and Pfizer; grants from Alcon Pharmaceuticals, Senju Pharmaceuticals, and Bayer Yakuhin; consultant's fee from Novartis Pharmaceuticals outside this work. Dr Kawashima received lecturer's fees from Kowa Pharmaceutical, Novartis Pharmaceuticals, and Santen Pharmaceuticals outside this work. Dr Yanagi received lecturer's fees and grants from Santen Pharmaceuticals outside this work. Dr Arai, Dr Takayama, Dr Yousefi, Dr Tampo, Dr Yamashita, Dr Hasegawa, Dr Ohgami, Dr Sonoda, Dr Tanaka, Dr Inoda, and Dr Sakamoto declare no potential conflict of interest. Hidenori Takahashi: Founder of DeepEyeVision LLC, outside this work. Yasuo Yanagi: Advisory board member for Bayer Pharmaceuticals. Consultant for Santen Pharmaceuticals. This does not alter our adherence to PLOS ONE policies on sharing data and materials.

central choroidal thickness (CCT) is associated with diverse ocular conditions, such as pachychoroid spectrum diseases including central serous chorioretinopathy [2], pachychoroid pigment epitheliopathy [3], pachychoroid neovasculopathy [4, 5], and polypoidal choroidal vasculopathy. Choroidal thickness can be measured only by OCT; however, this approach is not widely used owing to its high cost compared with that of colour fundus photography (CFP), which is commonly used in healthcare settings. Therefore, the accurate estimation of CCT from CFPs would be of potential benefit.

Deep learning, a branch of the evolving field of machine learning, has advanced substantially in recent years. Instead of humans specifying characteristic image parameters for identification, a deep learning system automatically extracts characteristic image features for learning based on a large number of images [6]. As a result, shapes and features that humans cannot recognise are automatically extracted, with high discriminatory ability. In 2014 and 2015, Google's deep convolutional neural network GoogLeNet [7] and Microsoft's deep convolutional neural network ResNet [8] exceeded the human limit for accuracy in image recognition.

In ophthalmology, many deep learning studies have focused on categorical classification, such as diabetic retinopathy disease staging and the age-related macular degeneration severity scale [9–12]. Deep learning is suitable for Euclidean data [13], but its application to ophthalmology is limited.

In this study, we developed a deep learning method to estimate choroidal thickness using normal central 45° field CFPs. The accuracy for unknown targets acquired under conditions different from those of the training dataset tends to be low [14]. Therefore, we estimated CCT using multicentre CFPs, discuss reasons for the decreased accuracy, and present a possible solution.

## Results

Measured CCT, axial length, age, and sex ratio for one or more validation institutions differed from those of institution A (Table 1).

We trained the GoogLeNet-like deep convolutional neural networks with 125, 250, 500, 1,000, and 2,548 45° posterior pole colour fundus photographs, using manually measured choroidal thickness obtained with a built-in measurement function of the OCT system. The standard errors (SEs) of the trained networks were 89, 79, 74, 65, and 73 μm, respectively (Fig 1). The SE showed a decreasing trend proportional to the amount of data, reaching a plateau when the number of images exceeded 500.

The images of the middle layer suggested the anatomical regions that might have been used by the newly developed estimation system. Choroidal vessels visible through the retinal pigment epithelium were observed (representative images are shown in Fig 2).

Although the deep-learning approach was relatively accurate, institution, age, sex, and axial length were significantly related to the estimation error, as expected. Accordingly, we re-estimated values corrected by camera, age, and axial length to improve the accuracy using multiple

**Table 1. Data for each institution.**

| Institution | Measured CCT (μm) | Axial length (mm) | Age (years) | Male (%) |
|---|---|---|---|---|
| A | 247 (115) | 23.9 (1.6) | 67 (14) | 62 |
| B | 306 (74, <0.0001) | 25.4 (1.4, <0.0001) | 25.8 (3.8, <0.0001) | 66 (0.3) |
| C | 245 (104, 1) | 24.3 (1.2, 0.02) | 56 (18, <0.0001) | 51 (0.008) |
| D | 339 (118, <0.0001) | 23.7 (1.1, 1) | 69 (15, 0.4) | 51 (0.03) |

Mean (standard deviation, $P$ vs. A), Steel test or $\chi^2$-test; CCT, central choroidal thickness.

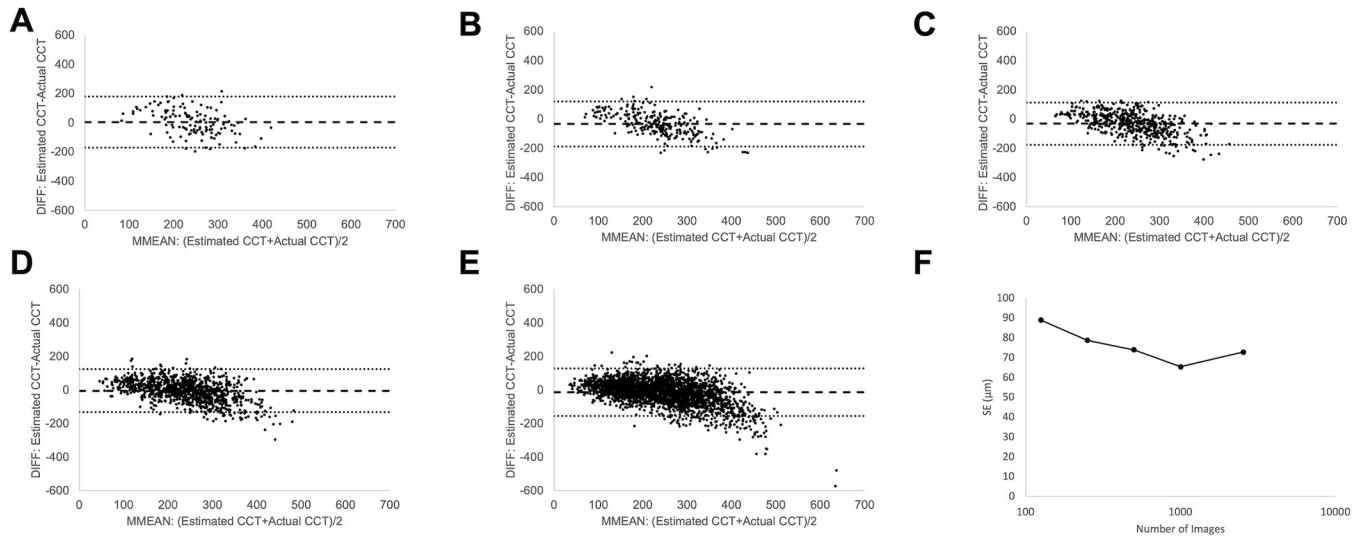

**Fig 1. Bland-Altman plots of choroid thickness estimated from fundus photographs.** A: 125, B: 250, C: 500, D: 1,000, and E: 2,548 images. Black dots: Estimated choroidal thickness from validation photographs. Horizontal axis: MMEAN, average of the actual and estimated thickness. Vertical axis: DIFF, difference between the actual and estimated thickness. Dotted line: limits of agreement; LOA, mean ± 1.96 standard error (SD) of the DIFF. F: Standard error (SE) of each estimation. SE decreased in proportion to the amount of data, reaching a plateau when the number of images exceeded 500.

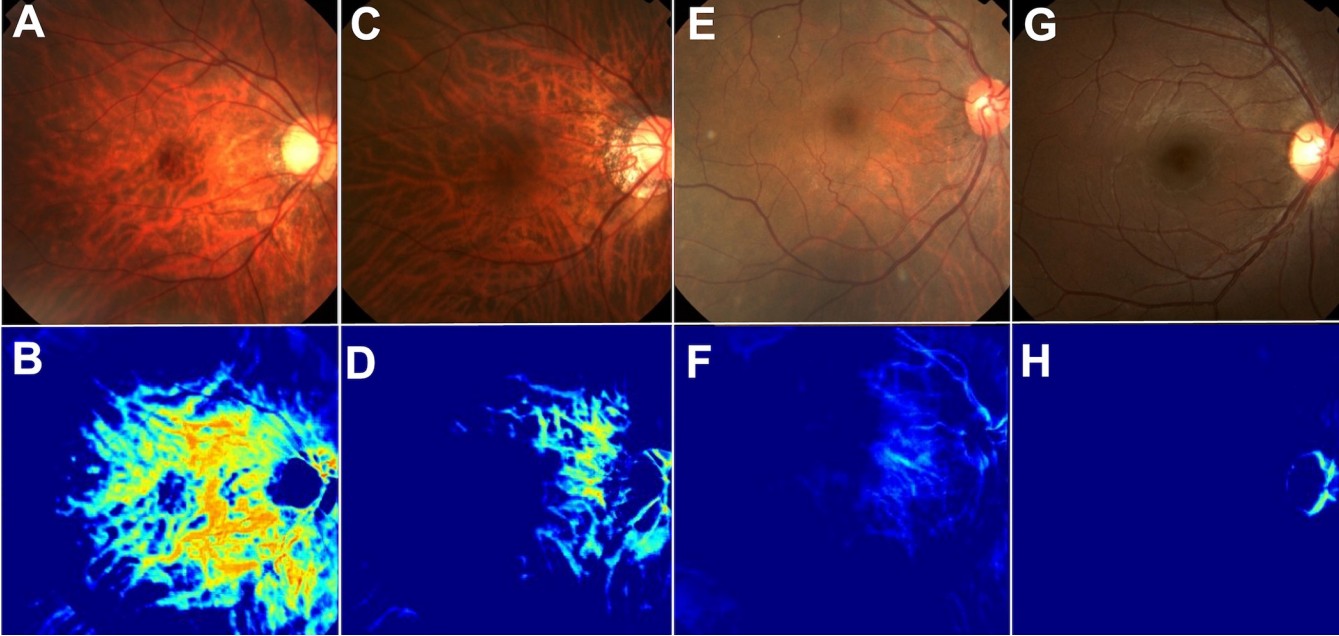

**Fig 2. Four representative colour fundus photographs measured CCT values, intermediate images, and estimated CCT values from colour fundus photographs.** A: Eye with CCT of 75 μm. The CCT estimated by the developed algorithm was 71 μm. B: The intermediate image was characterised by clear, larger-diameter choroidal vessels. C: Eye with CCT of 180 μm. The CCT estimated by the developed algorithm was 148 μm. D: The intermediate image was characterised by moderately clear, larger-diameter choroidal vessels. E: Eye with CCT of 266 μm. The CCT estimated by the developed algorithm was 304 μm. F: The intermediate image was characterised by unclear, larger-diameter choroidal vessels. G: Eye with CCT of 413 μm. The CCT estimated by the developed algorithm was 360 μm. H: The intermediate image did not show larger-diameter choroidal vessels.

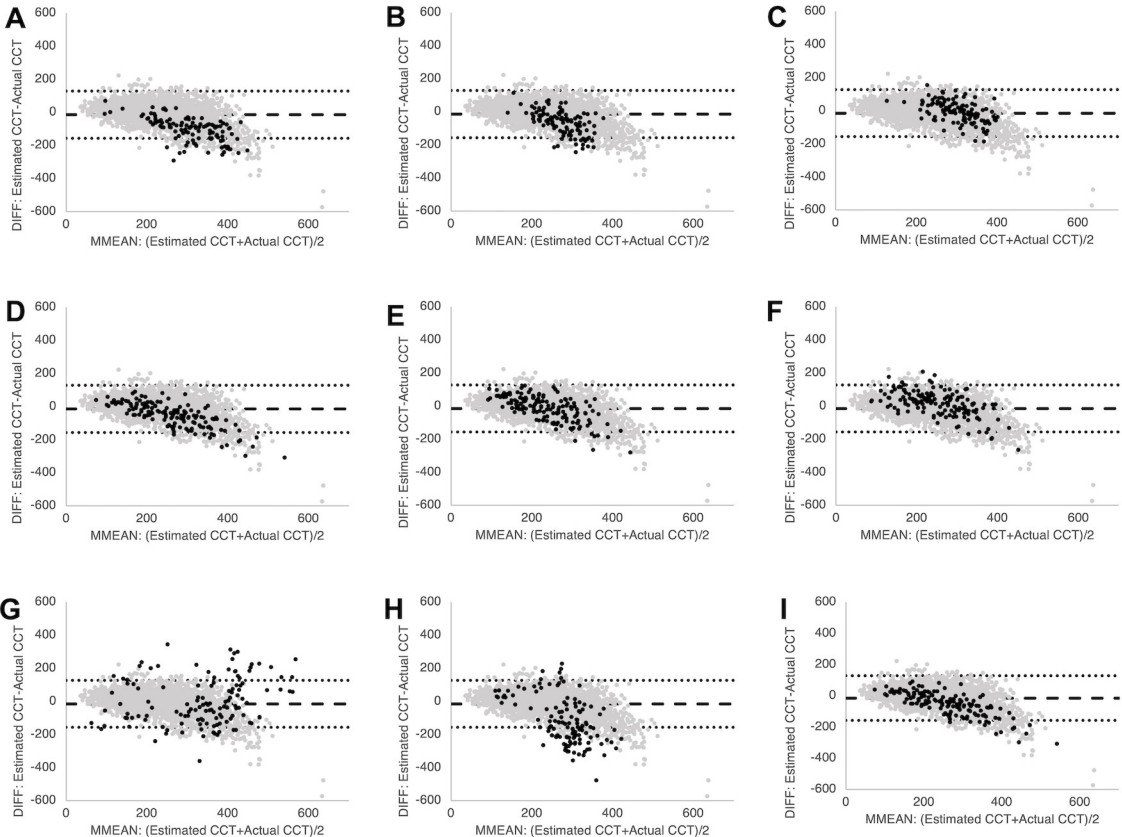

**Fig 3. Bland-Altman plots of central choroidal thickness (CCT) estimated from colour fundus photographs.** Horizontal axis: MMEAN, average of the actual and estimated thickness. Vertical axis: DIFF, difference between the actual and estimated thickness. Black dots: data from institution B, C, or D. Grey dots: data from institution A. Dotted line: limits of agreement; LOA, mean ± 1.96 standard error (SD) of the DIFF. A: At institution B, the estimated CCT was thinner than the actual CCT (mean error = -65 µm). Pearson's correlation coefficient $r = 0.31$. B: Corrected CCT was nearly the same as actual CCT (mean error = 3.4 µm) and $r$ significantly improved to 0.4 ($P = 0.001$ vs. A). C: After re-training, mean error = -6.8 µm and $r = 0.50$ ($P = 0.05$ vs. A). D: At institution C, mean error = -17 µm and $r = 0.68$. E: Corrected CCT was nearly the same as estimated CCT (mean error = 10 µm). Institution C used same camera as institution A. F: After re-training, mean error = 15 µm and $r = 0.67$ ($P = 0.9$ vs. D). G: At institution D, mean error = -120 µm and $r = 0.61$. H: Corrected CCT was thicker than the actual CCT (mean error = 40 µm) and $r$ improved to 0.62 ($P = 0.0001$ vs. G)). I: After re-training, mean error = -31 µm and $r = 0.55$ ($P = 0.4$ vs. G).

regression analysis (Fig 3A, 3B, 3D, 3E, 3G, and 3H). Moreover, we used data from all institutions for re-learning and found that the standard error for the retrained deep-learning was significantly improved (Fig 3A, 3C, 3D, 3F, 3G, and 3I) (Table 2). The standard error of CCT of sunset glow fundus was also low (Fig 4).

## Discussion

Ophthalmology is a main target of recent machine learning algorithms, but many challenges remain [9]. ter Haar Romeny et al. [10] reported a highly sophisticated brain-inspired algorithm for retinal image analysis. Using a very deep convolutional neural network, Xu et al. reported an accuracy of 94.5% [11]. In our previous study, we used diabetic disease staging information for unseen areas as training data for deep learning, achieving an accuracy of 80% [12]. Although the accuracy should be improved, the advantage of our previous network is two-fold: the use of single-field fundus photographs for diabetic retinopathy disease staging and the ease of screening fundus photographs.

**Table 2. Errors for original estimate and estimates after simple correction and after re-training.**

| Institution | Estimated CCT | Corrected CCT | After re-training |
|---|---|---|---|
| A | −14<br>74<br>0.78 | 1.6<br>70<br>0.78 (<0.0001) | −11 (<0.0001)<br>75 (na, 0.3)<br>0.77 (0.01) |
| B | −65<br>97 (<0.0001)<br>0.31 | 3.4<br>69 (0.2)<br>0.40 (0.001) | −6.8 (0.03)<br>65 (0.05, 0.1)<br>0.50 (0.05) |
| C | −17<br>80 (0.1)<br>0.68 | 10<br>76 (0.3)<br>0.69 (0.3) | 15 (0.3)<br>79 (0.2, 0.8)<br>0.67 (0.9) |
| D | −120<br>180 (<0.0001)<br>0.61 | 40<br>150 (<0.0001)<br>0.62 (0.0001) | −31 (<0.0001)<br>100 (<0.0001, 0.3)<br>0.55 (0.4) |

Each cell contains: Mean error (*P*-value for mean errors of corrected CCT vs. CCT after re-training (paired t-test)) (μm); standard error (*P*-value for standard errors vs. institution A (*F*-test) (μm), *P*-value for standard errors of corrected CCT vs. CCT after re-training (paired F-test)) (μm); Pearson's correlation coefficient *r* (*P*-value for *r* vs. Estimated CCT (paired Z-test)); CCT: central choroidal thickness.

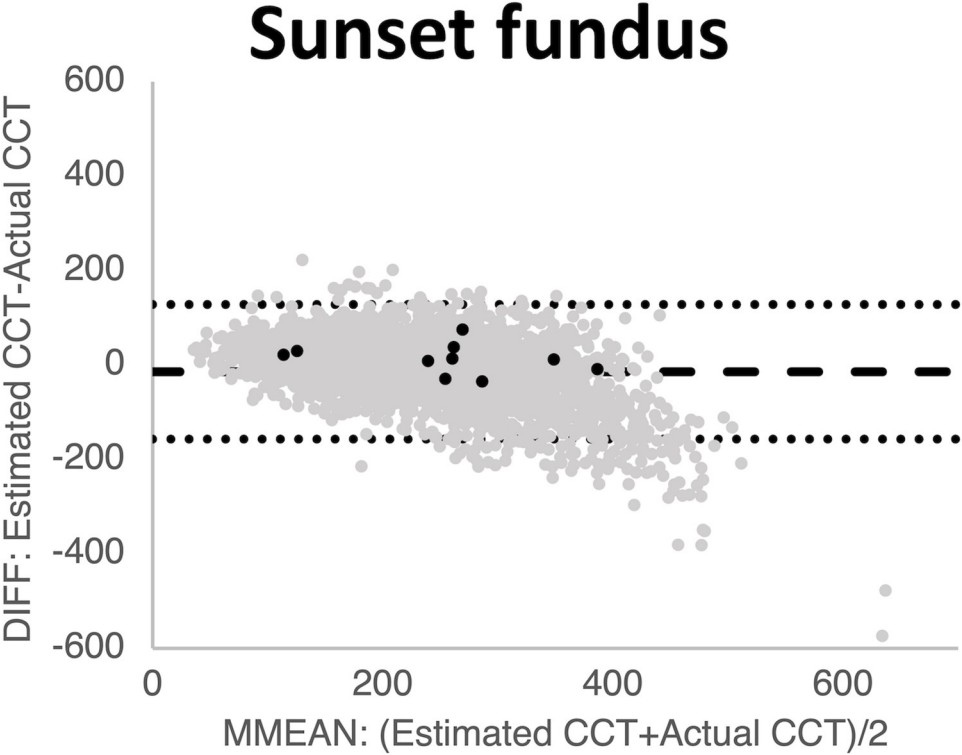

**Fig 4. Bland-Altman plots of central choroidal thickness (CCT) of 10 sunset-glow fundi estimated from colour fundus photographs.** Horizontal axis: MMEAN, average of the actual and estimated thickness. Vertical axis: DIFF, difference between the actual and estimated thickness. Black dots: data from 10 sunset fundus images. Grey dots: data from institution A. Dotted line: limits of agreement; LOA, mean ± 1.96 standard error (SD) of the DIFF. Estimated CCT was thinner than the actual CCT (mean error = 2.2 μm, standard error = 29 μm). Pearson's correlation coefficient *r* = 0.96.

Although the causality is not clear, several ocular conditions, such as central serous chorioretinopathy [2], pachychoroid neovasculopathy [4], polypoidal choroidal vasculopathy [15, 16], and primary open angle glaucoma [17], are associated with a thickened choroid. Therefore, it is rational to develop an algorithm to estimate choroidal thickness from fundus photographs. The present study clearly demonstrated that deep learning can estimate choroidal thickness with good accuracy. Although further studies are needed to validate the effectiveness our system for the assessment of fundus photographs, we believe that our algorithm is effective for identifying patients with a thickened choroid.

Several approaches have been developed to estimate choroidal thickness from fundus photographs. Komuku et al. (2020) used an adaptive binarization method to analyse choroidal vessels on colour fundus photographs and a deep-learning-based method to estimate the CCT based on the binarization-generated choroidal vessel images [18]. Li et al. (2021) used ResNet to estimate axial length and CCT based on colour fundus photographs [19]. Notably, both studies primarily relied on image processing for correction and did not address the correction of estimation errors. In contrast, our study has demonstrated that we can significantly enhance the accuracy of estimates by identifying factors associated with estimation errors, namely image quality, age, and axial length, and applying correction through multiple linear regression analysis. This finding is consistent with the results of a prior study indicating a strong correlation between CCT and axial length [20], enabling more precise predictions.

The convolutional neural network GoogLeNet was created for the general classification of $256 \times 256$ images [7], but the network is thought to be useful for the Euclidian classification of large images (e.g. $1,216 \times 1,216$ pixels, as in this experiment).

Deep-learning algorithms have been applied to data collected under various conditions, such as in single-institution studies [21], with a variety of cameras [22], and with a variety of races [23]. In this study, the correlation coefficient for measured CCT among different institutions using the same camera was low. Since the dataset from institution B included many fundus photographs of myopia in young people, correcting for the axial length improved the prediction accuracy. Although the datasets from institution D were similar to real-world data, the quality of CFPs was low for various reasons, including eccentricity of the fundus camera CCD mask, a dirty lens, poor patient fixation, out-of-focus, improper lens-to-pupil distance, and misalignment of camera optical axis and pupil centre. The particularly low quality of CFPs was thought to have contributed to the particularly low predictive accuracy of the CCT. Therefore, using only high-quality CFPs may be an easy way to increase accuracy. However, the algorithm created with only good-quality CFPs may have a low accuracy for CFPs obtained by different cameras, hindering its general use. To improve robustness, re-training with a relatively small set of target data may be useful, as shown in the present study.

The distinctive red hue of the sunset glow fundus (SGF) introduces a potential margin of error in the inference of choroidal thickness. This condition was identified through a search based on the diagnosis recorded in medical records, further validated by fundus photographs and patient history. The definition of the SGF is a funduscopic finding characterized by orange-red discoloration due to depigmentation of the inflamed choroid [24]. The sunset glow fundus appears different, but the standard error was low. It is possible that choroidal vessels are easy to see in this case.

This study had some limitations. First, scatter plots showed that the thicker the choroid, the lower the estimation accuracy. It is challenging to estimate choroidal thickness for those >400 µm because it is difficult to see the choroidal vessels in eyes. Other neural networks may achieve higher accuracy. Second, our analysis was based on conventional one-field 45˚ fundus photographs. Machine learning algorithms can be trained on more parameters, including sex, age, and race, and this may lead to greater accuracy.

However, with newer neural network systems, we expect that the current system would achieve a clinically useful accuracy. Our proposed deep learning choroidal thickness estimation system can predict the risk of central serous chorioretinopathy [2] and the onset of pachy-choroid neovasculopathy [4]. We believe that this estimation system has the potential to help graders judge choroidal thickness and to promote improved disease screening.

## Conclusions

We proposed a novel algorithm based on deep learning that estimates choroidal thickness using only colour fundus photographs. The accuracy of deep learning improved with values corrected for multiple regression analysis using camera, age, and axial length and additional training with small data sets from other institutions. This algorithm has the potential to help graders judge choroidal thickening and thinning.

## Materials and methods

### Study design

This multiple-site, retrospective, exploratory study was performed in an institutional setting.

### IRB approval

Institutional review board approval was obtained from the Jichi Medical University Research Ethics Committee [CU22-140]. For this study, solely de-identified retrospective clinical data were utilized. Given the nature of the data, the Institutional Review Board (IRB) granted an opt-out option and waived the requirement for informed consent. The protocol adhered to the tenets of the Declaration of Helsinki.

### Subjects

In total, 2,548 posterior pole photographs were obtained from 1166 eyes of 775 patients who underwent same-day swept-source OCT examination (DRI-OCT Atlantis; Topcon Co., Ltd., Tokyo, Japan) and CFP (VX-10; Kowa Co., Ltd., Nagoya, Japan) at the Jichi Medical University Hospital outpatient clinic (institution A) between 2014 and 2017. When more than six images were available for a single eye, only five were used.

A validation study was performed using data from Kagoshima University (B), Saitama Medical Center (C), and Ibaraki Seinan Medical Center (D), obtained using various cameras (Table 3). Additionally, 10 sunset glow fundus images were used for validation.

Institution A is a university hospital with a high incidence of critical patients, yet data also include images of their healthy eyes. Facility B's data pertain to students undergoing clinical

**Table 3. Cameras in use at each institution.**

| Institution | Camera | OCT | n |
| --- | --- | --- | --- |
| A | Kowa VX-10 | TOPCON DRI-OCT Atlantis | 2,548 |
| B | TOPCON TRC-50LX<br>TOPCON TRC-NW7S | Heidelberg SPECTRALIS OCT | 115 |
| C | Kowa VX-10<br>TOPCON DRI-OCT Triton | TOPCON DRI-OCT Triton | 148 |
| D | TOPCON 3D OCT-2000 | TOPCON 3D OCT-2000 | 129 |

OCT, optical coherence tomography; Kowa Company, Ltd., Nagoya, Japan; Topcon Corporation, Tokyo, Japan; Heidelberg Engineering GmbH, Heidelberg, Germany.

training. Facility C represents a general hospital located in an urban center, whereas Facility D is a general hospital situated in a rural area.

The data were accessed for research purposes from February 22, 2016, to September 30, 2023. The authors did not have access to information that could identify individual participants during or after data collection.

## Images

All subjects were imaged between 9 AM and 4 PM, 30 minutes after mydriasis. Mydriatic 45˚ field central colour fundus photographs were obtained. This study was retrospective in nature, leading to inconsistencies in the OCT scan modes utilized. Within the same facility, the type of examination varied; general clinics conducted 6 mm radial scans with six lines, whereas macular specialty clinics performed 12 mm radial scans with twelve lines. CCT was measured manually using a built-in measurement function of the OCT system. Choroidal thickness measurements were conducted blindly by a single optometrist (H. Tampo) with 20 years of experience. In spectral-domain OCT (SD-OCT), when the choroid was exceptionally thick, delineation of the boundary with the sclera was often unclear. However, an effort was made to report the most accurate values possible.

Exclusion criteria were any signs of apparent macular conditions, such as subretinal haemorrhage, central serous detachment, retinal pigment epithelial detachment, and macular oedema. All other fundus photographs of good quality were included. Most of the fundus photographs were of normal fellow eyes of the outpatients. Eyes with moderate retinal haemorrhage, drusen, moderate exudates, and glaucoma were included.

Multiple imaging sessions for the same patient were employed; however, a limit was set to a maximum of five visits per patient to prevent an excessive accumulation of images from any single individual. Additionally, due to the re-imaging of the lowest 5% in terms of quality, instances occurred where the same eye was captured in two photographs during a single visit.

The filenames of the fundus photographs initially contained patient IDs and the dates and times of capture. To ensure patient confidentiality, the ID portion was replaced with a sequential numbering system, and the capture dates were randomly shifted within a one-week interval. Furthermore, to achieve image standardization, each photograph was processed to remove any surrounding margins, converting it into a square shape, followed by resizing to ensure uniformity in size across all images. These steps facilitated the normalization of the dataset for subsequent analysis.

## Model development

The original photographs were 1,604 × 1,216 pixels. The central areas in 1,216 × 1,216 pixels were used. For 10-fold cross-validation, 2,548 fundus photographs were divided into 10 groups. Photographs of the same patient were included in the same group. In each training protocol, 2,293 fundus photographs were used for training and the remaining 255 fundus photographs were used for validation. The number of training epochs was the point at which the loss of validation in the first training/validation pair dropped sufficiently and not the epochs where the loss was coincidentally low. No overfitting was observed. The same number of epochs were used to train the remaining nine training/validation pairs. All 10 pairs were included in the validation set and CCT was calculated. The 10-fold cross-validation was also performed on 125, 250, 500 and, 1,000 randomly chosen images. Each sampling was performed five times and the average thickness was used. The number of training epochs was same as the number of epochs used in the total dataset described above.

A fully randomly initialized GoogLeNet-like deep convolutional neural network was used because ResNet-like deep convolutional neural networks did not have high accuracy using our dataset. A grid search from $10^{-3}$ to $10^{3}$ or from $2^{-3}$ to $2^{3}$ for each hyperparameter was used for ResNet and GoogLeNet. The following GoogLeNet modifications were applied: Softmax with Loss Layer was changed to the Euclidean Loss Layer, number of variables from the layer used was changed from 1,024 to 4, deletion of the top 5 accuracy layers, expansion of the crop size to 1,064 pixels, reduction of the batch size to 24, and the base learning rate set to $1e^{-7}$. The neural network was used in an open framework for deep learning Caffe (Berkeley Vision and Learning Center, Berkeley, CA, USA). The neural networks were trained for 3,000 epochs.

Four graphical processing units (Quadro P6000; NVIDIA Co., Santa Clara, CA, USA) were used simultaneously. The deep learning system was run on a workstation for 82 h at each training.

## Statistical analysis

The standard error for 2,548 fundus photographs used as validation was calculated as a main outcome. Multiple regression analysis was used after stepwise variable selection to evaluate the relationship between estimation error and various parameters, such as institution, camera, OCT camera, age, sex, and axial length. Statistical analyses were performed using JMP version 14.1.0 (SAS Institute, Cary, NC, USA).

In four randomly chosen validation images, characteristics used by the newly developed algorithm in middle-layer images were evaluated.

## Supporting information

**S1 File. Approval document Japanese.**
(DOCX)

**S2 File. Approval document English translated.**
(DOCX)

## Author Contributions

**Conceptualization:** Hidetoshi Kawashima, Yasuo Yanagi.

**Data curation:** Hironobu Tampo, Tomohiro Ohgami, Shozo Sonoda, Yoshiaki Tanaka.

**Formal analysis:** Siamak Yousefi, Yoshiaki Tanaka.

**Methodology:** Hidenori Takahashi.

**Resources:** Takehiro Yamashita, Satoru Inoda, Shinichi Sakamoto.

**Supervision:** Hidenori Takahashi, Hidetoshi Kawashima, Yasuo Yanagi.

**Validation:** Tetsuya Hasegawa.

**Writing – original draft:** Yusuke Arai.

**Writing – review & editing:** Takuya Takayama.

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
