## [Decision Letter · Decision Letter 0]

4 Jan 2024

PONE-D-23-35252Predicting central choroidal thickness from colour fundus photographs using deep learningPLOS ONE

Dear Dr. Takahashi,

Thank you for submitting your manuscript to PLOS ONE. After careful consideration, we feel that it has merit but does not fully meet PLOS ONE’s publication criteria as it currently stands. Therefore, we invite you to submit a revised version of the manuscript that addresses the points raised during the review process.

We look forward to receiving your revised manuscript.

Kind regards,

Tatsuya Inoue

Academic Editor

PLOS ONE

“Dr Arai, Dr Takayama, Dr Yousefi, Dr Tampo, Dr Yamashita, Dr Hasegawa, Dr Ohgami, Dr Sonoda, Dr Tanaka, Dr Inoda, and Dr Sakamoto declare no potential conflict of interest. Dr Takahashi: Lecturer’s fees from Novartis Pharmaceuticals, Kowa Pharmaceutical, Senju Pharmaceutical, Alcon Pharmaceuticals, Santen Pharmaceutical, and Pfizer; grants from Alcon Pharmaceuticals, Senju Pharmaceuticals, and Bayer Yakuhin; consultant’s fee from Novartis Pharmaceuticals outside this work. Dr Kawashima received lecturer’s fees from Kowa Pharmaceutical, Novartis Pharmaceuticals, and Santen Pharmaceuticals outside this work. Dr Yanagi received lecturer’s fees and grants from Santen Pharmaceuticals outside this work. Dr Arai, Dr Takayama, Dr Yousefi, Dr Tampo, Dr Yamashita, Dr Hasegawa, Dr Ohgami, Dr Sonoda, Dr Tanaka, Dr Inoda, and Dr Sakamoto declare no potential conflict of interest. Hidenori Takahashi: Founder of DeepEyeVision LLC, outside this work. Yasuo Yanagi: Advisory board member for Bayer Pharmaceuticals. Consultant for Santen Pharmaceuticals.”

Reviewers' comments:

Reviewer's Responses to Questions

**Comments to the Author**

1. Is the manuscript technically sound, and do the data support the conclusions?

Reviewer #1: Partly

Reviewer #2: Yes

2. Has the statistical analysis been performed appropriately and rigorously? 

Reviewer #1: Yes

Reviewer #2: Yes

3. Have the authors made all data underlying the findings in their manuscript fully available?

Reviewer #1: Yes

Reviewer #2: Yes

4. Is the manuscript presented in an intelligible fashion and written in standard English?

Reviewer #1: Yes

Reviewer #2: Yes

5. Review Comments to the Author

Reviewer #1: 1.Given that choroidal thickness is dynamic, details regarding the OCT scanning mode, image acquisition timing, and specific measurement methods for choroidal thickness should all be provided.

2.The number of fundus photos for the study is adequate, but there is a relatively small number of patients. Multiple photos have been obtained from one eye; what are the differences among these photos? Specifically, differing capture angles might affect the training efficacy of the model，such as it could lead to overfitting of the model.please provide a specific explanation of this issue.

3.The convolutional neural network is a black-box process, and its ability to learn image features can be easily influenced by irrelevant factors. Is it possible to conduct SHAP analysis on the model training results and generate heatmaps to explicitly identify the learned features of the model?"

4.How is the consistency of image quality maintained from multiple machines included in the study? Please provide a detailed description of the pre-processing workflow for fundus images and the information de-identification process.

5.Previously, there have been studies that developed artificial intelligence systems using fundus images to predict choroidal thickness, and their sample sizes were larger than the current study. Therefore, the innovation of this study is limited.

6.The selection of the study population has a significant impact on the specificity of the model. Could you provide a detailed description of the ocular and systemic parameters of the population included in this study?

7.In the results, metrics such as sensitivity, specificity, accuracy, ROC curves, etc., should be used to demonstrate the performance of the model.

Reviewer #2: Many thanks for the opportunity to review this manuscript entitled " Predicting central choroidal thickness from colour fundus photographs using deep learning ". While this manuscript is interesting, I take this opportunity to comment on several issues.

・Page 3-4, An increase in central choroidal thickness (CCT) is associated with diverse ocular conditions, such as pachychoroid spectrum diseases including central serous chorioretinopathy, pathychoroid pigment epitheliopathy, pachychoroid neovasculopathy, and as well as other ocular diseases, such as glaucoma. According to the cited references, choroids were significantly (approximately 50 microns) thinner in glaucoma than in normal. Please correct.

・It is difficult to understand what each number in Table 2 means. Please write clearly what each number represents.

・Figure caption of Fig 1 and Fig 3, make lowercase letters uppercase.

・Page 6, The description of sunset glow fundus is insufficient. Please add a description of the definition of sunset glow fundus, how you selected cases, and why you considered cases with sunset glow fundus.

6. PLOS authors have the option to publish the peer review history of their article (what does this mean?). If published, this will include your full peer review and any attached files.

Reviewer #1: No

Reviewer #2: No

---

## [Author Response · Author response to Decision Letter 0]

13 Feb 2024

We wish to express our appreciation to the reviewer for your insightful comments, which have helped us significantly improve the paper.

Comment 1: 1. Please ensure that your manuscript meets PLOS ONE's style requirements, including those for file naming. The PLOS ONE style templates can be found at https://journals.plos.org/plosone/s/file?id=wjVg/PLOSOne_formatting_sample_main_body.pdf

and 

Response: We have revised our manuscript following "The PLOS ONE style templates."

Comment 2: 2. Thank you for stating the following in the Competing Interests section:

“Dr Arai, Dr Takayama, Dr Yousefi, Dr Tampo, Dr Yamashita, Dr Hasegawa, Dr Ohgami, Dr Sonoda, Dr Tanaka, Dr Inoda, and Dr Sakamoto declare no potential conflict of interest. Dr Takahashi: Lecturer’s fees from Novartis Pharmaceuticals, Kowa Pharmaceutical, Senju Pharmaceutical, Alcon Pharmaceuticals, Santen Pharmaceutical, and Pfizer; grants from Alcon Pharmaceuticals, Senju Pharmaceuticals, and Bayer Yakuhin; consultant’s fee from Novartis Pharmaceuticals outside this work. Dr Kawashima received lecturer’s fees from Kowa Pharmaceutical, Novartis Pharmaceuticals, and Santen Pharmaceuticals outside this work. Dr Yanagi received lecturer’s fees and grants from Santen Pharmaceuticals outside this work. Dr Arai, Dr Takayama, Dr Yousefi, Dr Tampo, Dr Yamashita, Dr Hasegawa, Dr Ohgami, Dr Sonoda, Dr Tanaka, Dr Inoda, and Dr Sakamoto declare no potential conflict of

interest. Hidenori Takahashi: Founder of DeepEyeVision LLC, outside this work. Yasuo Yanagi: Advisory board member for Bayer Pharmaceuticals. Consultant for Santen Pharmaceuticals.”

Please confirm that this does not alter your adherence to all PLOS ONE policies on sharing data and materials, by including the following statement: "This does not alter our adherence to PLOS ONE policies on sharing data and materials.” (as detailed online in our guide for authors http://journals.plos.org/plosone/s/competing-interests). If

there are restrictions on sharing of data and/or materials, please state these. Please note that we cannot proceed with consideration of your article until this information has been declared.

Response: To confirm our continued adherence to all PLOS ONE policies on sharing data and materials, we have included the following statement: "This does not alter our adherence to PLOS ONE policies on sharing data and materials.” We have included our updated Competing Interests statement in our cover letter, and we appreciate your assistance in changing the online submission form on our behalf. 

RESPONSE TO REVIEWER 1

We wish to express our appreciation to the reviewer for your insightful comments, which have helped us significantly improve the paper.

Comment 1: Given that choroidal thickness is dynamic, details regarding the OCT scanning mode, image acquisition timing, and specific measurement methods for choroidal thickness should all be provided.

Response: We thank the reviewer for this comment.

We have added “All subjects were imaged between 9 AM and 4 PM, 30 minutes after mydriasis. This study was retrospective in nature, leading to inconsistencies in the OCT scan modes utilized. Within the same facility, the type of examination varied; general clinics conducted 6 mm radial scans with six lines, whereas macular specialty clinics performed 12 mm radial scans with twelve lines. Choroidal thickness measurements were conducted blindly by a single optometrist (H. Tampo) with 20 years of experience. In spectral-domain OCT (SD-OCT), when the choroid was exceptionally thick, delineation of the boundary with the sclera was often unclear. However, an effort was made to report the most accurate values possible.” to the method section. 

Comment 2: The number of fundus photos for the study is adequate, but there is a relatively small number of patients. Multiple photos have been obtained from one eye; what are the differences among these photos? Specifically, differing capture angles might affect the training efficacy of the model, such as it could lead to overfitting of the model. Please provide a specific explanation of this issue.

Response: We thank the reviewer for this pertinent comment.

In response to concerns raised, multiple imaging sessions per patient were adopted to augment the dataset size given the limited number of patients available at the facility. This approach could potentially influence the training effectiveness of the model. To mitigate the risk of overfitting, a restriction was implemented whereby no single patient's imaging sessions would exceed five visits. Furthermore, to ensure quality control, the bottom 5% of images in terms of quality were re-imaged, leading to instances where the same eye was photographed twice in a single visit. These methodologies were duly documented in the methods section of our study.

Comment 3: The convolutional neural network is a black-box process, and its ability to learn image features can be easily influenced by irrelevant factors. Is it possible to conduct SHAP analysis on the model training results and generate heatmaps to explicitly identify the learned features of the model?"

Response: We thank the reviewer for this comment.

The framework utilized in this study, Caffe, has not been maintained, rendering the reevaluation of the AI components exceedingly challenging. We regret to inform that the implementation of SHAP analysis could not be carried out in this study. We acknowledge the importance of SHAP for providing interpretable machine learning insights and its potential to enhance the understanding of model predictions. Despite our best efforts, limitations above have precluded us from integrating this analysis. We appreciate the reviewers' understanding.

Comment 4: How is the consistency of image quality maintained from multiple machines included in the study? Please provide a detailed description of the pre-processing workflow for fundus images and the information de-identification process.

Response: We thank the reviewer for this comment.

In response to the reviewers' concerns regarding the identification of patients from fundus photograph filenames, we have implemented several measures to ensure anonymity. Specifically, we replaced the ID portion of each filename with a sequential number and adjusted the date portion randomly within a one-week range to prevent patient identification. Furthermore, to standardize the images, we removed any margins, converted the images to a square format, and then resized them to match in size, thus achieving normalization. These modifications have been detailed in the methods section of our manuscript to provide clarity on our approach to maintaining patient confidentiality and image standardization.

Comment 5: Previously, there have been studies that developed artificial intelligence systems using fundus images to predict choroidal thickness, and their sample sizes were larger than the current study. Therefore, the innovation of this study is limited.

Response: We thank the reviewer for this comment. As you pointed out, previous studies had larger sample sizes than our research. However, as discussed in the Discussion section, those studies did not address the correction of estimation errors. There are several methods to enhance the accuracy of estimates, and increasing the sample size is one potential solution. However, for estimating CCT from fundus photographs, numerous relevant factors, including image quality, age, and axial length, come into play. Our study is innovative in identifying many of these factors and significantly improving estimation accuracy by applying corrections through multiple linear regression analysis.

Comment 6: The selection of the study population has a significant impact on the specificity of the model. Could you provide a detailed description of the ocular and systemic parameters of the population included in this study?

Response: We thank the reviewer for this comment.

Apart from the data specified, including age, gender, and axial length, no further information was collected. However, the characteristics of each institution and the subjects have been detailed as follows in the Subject section of our manuscript. “Facility A is a university hospital with a high incidence of critical patients, yet data also include images of healthy eyes. Facility B's data pertain to students undergoing clinical training. Facility C represents a general hospital located in an urban center, whereas Facility D is a general hospital situated in a rural area.”

Comment 7: In the results, metrics such as sensitivity, specificity, accuracy, ROC curves, etc., should be used to demonstrate the performance of the model.

Response: We thank the reviewer for this comment.

In response to the inquiry regarding the calculation of sensitivity and other statistical measures, it is pertinent to note that our analysis involves continuous variables, rendering the computation of sensitivity infeasible without a clear threshold. Although a distinct threshold for pachychoroid would allow for the determination of sensitivity, the lack of a universally established threshold—owing to the natural decline of choroidal thickness with age—complicates such calculations. Consequently, our study does not present specific sensitivity values due to these methodological constraints.

 

RESPONSE TO REVIEWER 2

We wish to express our appreciation to the reviewer for your insightful comments, which have helped us significantly improve the paper.

Comment 1: Page 3-4, An increase in central choroidal thickness (CCT) is associated with diverse ocular conditions, such as pachychoroid spectrum diseases including central serous chorioretinopathy, pathychoroid pigment epitheliopathy, pachychoroid neovasculopathy, and as well as other ocular diseases, such as glaucoma. According to the cited references, choroids were significantly (approximately 50 microns) thinner in glaucoma than in normal. Please correct.

Response: We thank the reviewer for this comment. I have removed the text and references related to glaucoma from the manuscript.

Comment 2: It is difficult to understand what each number in Table 2 means. Please write clearly what each number represents.

Response: We thank the reviewer for this pertinent comment.

In response to the comments received, it has been noted that the explanation provided beneath the table could benefit from further clarification to ensure its intent as descriptive text is immediately apparent. To address this, we have prefaced the explanation with "Each cell contains;" to clearly denote that the subsequent text serves as an explanation for the contents of each cell in the table.

Comment 3: Figure caption of Fig 1 and Fig 3, make lowercase letters uppercase.

Response: We thank the reviewer for this comment. We have modified the lowercase letters to uppercase as suggested.

Comment 4: Page 6, The description of sunset glow fundus is insufficient. Please add a description of the definition of sunset glow fundus, how you selected cases, and why you considered cases with sunset glow fundus.

Response: We thank the reviewer for this comment. We have added “The distinctive red hue of the sunset glow fundus (SGF) introduces a potential margin of error in the inference of choroidal thickness. This condition was identified through a search based on the diagnosis recorded in medical records, further validated by fundus photographs and patient history. The definition of the SGF is a funduscopic finding characterized by orange-red discoloration due to depigmentation of the inflamed choroid.” to the discussion section.

---

## [Decision Letter · Decision Letter 1]

18 Mar 2024

Predicting central choroidal thickness from colour fundus photographs using deep learning

PONE-D-23-35252R1

Dear Dr. Takahashi,

We’re pleased to inform you that your manuscript has been judged scientifically suitable for publication and will be formally accepted for publication once it meets all outstanding technical requirements.

Kind regards,

Tatsuya Inoue

Academic Editor

PLOS ONE

Additional Editor Comments (optional):

Reviewers' comments:

Reviewer's Responses to Questions

**Comments to the Author**

1. If the authors have adequately addressed your comments raised in a previous round of review and you feel that this manuscript is now acceptable for publication, you may indicate that here to bypass the “Comments to the Author” section, enter your conflict of interest statement in the “Confidential to Editor” section, and submit your "Accept" recommendation.

Reviewer #2: All comments have been addressed

2. Is the manuscript technically sound, and do the data support the conclusions?

Reviewer #2: Yes

3. Has the statistical analysis been performed appropriately and rigorously? 

Reviewer #2: Yes

4. Have the authors made all data underlying the findings in their manuscript fully available?

Reviewer #2: Yes

5. Is the manuscript presented in an intelligible fashion and written in standard English?

Reviewer #2: Yes

6. Review Comments to the Author

Reviewer #2: (No Response)

7. PLOS authors have the option to publish the peer review history of their article (what does this mean?). If published, this will include your full peer review and any attached files.

Reviewer #2: No

---

## [Editor Report · Acceptance letter]

20 Mar 2024

PONE-D-23-35252R1 

PLOS ONE

Dear Dr. Takahashi, 

I'm pleased to inform you that your manuscript has been deemed suitable for publication in PLOS ONE. Congratulations! Your manuscript is now being handed over to our production team.

Kind regards, 

on behalf of

Dr. Tatsuya Inoue 

Academic Editor

PLOS ONE